# Advances in Development of Novel Therapeutic Strategies against Multi-Drug Resistant *Pseudomonas aeruginosa*

**DOI:** 10.3390/antibiotics13020119

**Published:** 2024-01-25

**Authors:** Changhong Yin, Md Zahidul Alam, John T. Fallon, Weihua Huang

**Affiliations:** Department of Pathology and Laboratory Medicine, Brody School of Medicine, East Carolina University, Greenville, NC 27834, USA; alamm23@ecu.edu (M.Z.A.); fallonj19@ecu.edu (J.T.F.); huangwe21@ecu.edu (W.H.)

**Keywords:** *Pseudomonas aeruginosa*, lipopolysaccharide, porin, multi-drug resistance, phage therapy

## Abstract

*Pseudomonas aeruginosa* (*P. aeruginosa*) with multi-drug resistance (MDR) is a major cause of serious healthcare-associated infections, leading to high morbidity and mortality. This opportunistic pathogen is responsible for various infectious diseases, such as those seen in cystic fibrosis, ventilator-associated pneumonia, urinary tract infection, otitis externa, and burn and wound injuries. Due to its relatively large genome, *P. aeruginosa* has great diversity and can use various molecular mechanisms for antimicrobial resistance. For example, outer membrane permeability can contribute to antimicrobial resistance and is determined by lipopolysaccharide (LPS) and porin proteins. Recent findings on the regulatory interaction between peptidoglycan and LPS synthesis provide additional clues against pathogenic *P. aeruginosa*. This review focuses on recent advances in antimicrobial agents and inhibitors targeting LPS and porin proteins. In addition, we explore current and emerging treatment strategies for MDR *P. aeruginosa*, including phages, vaccines, nanoparticles, and their combinatorial therapies. Novel strategies and their corresponding therapeutic agents are urgently needed for combating MDR pathogens.

## 1. Introduction

*Pseudomonas aeruginosa* (*P. aeruginosa*) is a common opportunistic human pathogen. It often causes various complicated acute and chronic infections in immunocompromised hosts. *P. aeruginosa* can multiply and become the main pathogen in cystic fibrosis (CF) patients, ventilator-associated pneumonia, urinary tract infection, otitis externa, burn and wound injuries, bone and joint infections, and systemic infections. Antimicrobial resistance (AMR) is an urgent global public health threat resulting in the death of at least 1.27 million people worldwide and was associated with nearly 5 million deaths in 2019 [1]. In the US alone, AMR underlines 2.8 million infections and 35,000 deaths per year according to the Centers for Diseases Control and Prevention (CDC)’s 2019 Antibiotic Resistance Threats Report. Multidrug-resistant (MDR) and extensively drug-resistant (XDR) *P. aeruginosa* isolates are frequent causes of serious nosocomial infections and major sources of morbidity and mortality. Data from the CDC show that 8.9% of *P. aeruginosa* isolates were MDR in 2021 [2]. A detailed report from the National Healthcare Safety Network shows that 18.6% of MDR isolates were from intensive care unit patients, 29.9% from long-term acute-care hospitals, and 11.6% from hospital oncology units [3]. *P. aeruginosa* is one of the six MDR ESKAPE pathogens including *Enterococcus faecium, Staphylococcus aureus, Klebsiella pneumoniae, Acinetobacter baumannii*, and *Enterobacter* spp., which cause life-threatening nosocomial infections. In 2017, the World Health Organization (WHO) listed carbapenem-resistant *P. aeruginosa* as a critical-priority bacterium requiring the urgent development of new antimicrobials to counter a growing global public health crisis [4]. During the COVID-19 pandemic, *P. aeruginosa* was identified as the second most common bacterial co-infection in patients with COVID-19 [5]. The occurrence of severe infections by MDR *P. aeruginosa* further increased the complexity of the clinical management of COVID-19 patients [5,6]. In January 2023, an outbreak of a rare strain of XDR *P. aeruginosa* linked to eye drops was reported by the CDC. As of 15 May 2023, 81 patients were identified in the US as part of the outbreak [7]. The outbreak involved a carbapenem-resistant *P. aeruginosa* strain carrying the Verona Integron-encoded Metallo-beta-lactamase (VIM) and Guiana Extended-Spectrum beta-lactamase (GES) genes.

The molecular mechanisms of AMR are complex. The continuous use of various antibiotics over the years has led bacteria to accumulate various AMR mechanisms [8,9]. AMR in *P*. *aeruginosa* can be caused by low outermembrane (OM) permeability, drug-resistant efflux pumps, the presence of antibioticresistance genes, the formation of biofilms, etc. Noticeably, multiple mechanisms are simultaneously present, resulting in resistance to nearly all antibiotics available against *P. aeruginosa* [10]. AMR mostly depends on the structure and composition of the bacterial cell surface, especially alterations in the OM of Gram-negative bacteria. The OM acts as a frontline defense against hostile environments. The XDR *P. aeruginosa* is highly related to low OM permeability. The characterization of the OM is essential for understanding how antibiotics penetrate this barrier, and for the subsequent development of new therapeutic strategies. In this review, we focus on the major pathogenic components of LPS and porins, which play a significant role in OM permeation, and their relevant therapeutic strategies in development against MDR *P. aeruginosa* (Figure 1).

## 2. Mechanisms of Antimicrobial Resistance Targeting LPS and Porins

*P. aeruginosa* has an asymmetric OM composed of phospholipid in the inner leaflet and lipopolysaccharide (LPS) glycolipid in the outer leaflet, compared to a cytoplasmic inner membrane (IM) with a symmetric phospholipid bilayer. LPS has received much attention because of its ability to stimulate the host immune system as an endotoxin [11]. It is now known that OM provides a highly selective permeability barrier against many toxic compounds such as host antimicrobial peptides and antibiotics, as well as both lipophilic and hydrophilic compounds, including nutrients. The high selectivity of the OM in *P. aeruginosa* is mainly attributed to the presence of LPS [12].

### 2.1. LPS Biosynthesis

LPS is a major surface molecule of Gram-negative bacteria and consists of three distinct domains: lipid A, the hydrophobic portion of LPS that anchors the molecule in the OM; core oligosaccharide; and O antigen, an extended polysaccharide chain extending into the extracellular environment. Lipid A is essential for bacteria growth. Mutants with reduced lipid A biosynthesis grow slowly and are sensitive to a wide range of antibiotics [13]. The absence of lipid A impedes the aggregation of LPS, leading to bacteria cell death [14]. The biosynthesis of lipid A relies on a zinc-dependent metalloamidase, UDP-3-O-(R-3-hydroxymyristoyl)-N-acetylglucosamine deacetylase (*LpxC*). Owing to the critical role of *LpxC* in lipid A biosynthesis and its lack of homology with mammalian proteins, *LpxC* inhibitors are expected to be potential antibiotics for the treatment of Gram-negative bacterial infections [15,16,17].

An extensive research effort has focused on the discovery of novel *LpxC* inhibitors against *P. aeruginosa*. CHIR-090 was the first reported *LpxC* inhibitor in 2005 [18]. Another two compounds, ACHN-975 [19,20] and PF-5081090 [21] (Figure 1 and Table 1), discovered later, exhibit extensive antimicrobial activity. These two *LpxC* inhibitors are more active against *P. aeruginosa* with lower minimal inhibitory concentration MIC_90_ (0.5~1 μg/mL) and IC_50_ (1.1 nM) than CHIR-090 [22,23]. However, few inhibitors of *LpxC* have reached clinical trials, largely due to the limited efficacy and unfavorable cardiovascular toxicity of the candidate inhibitors tested [24]. ACHN-975 is the first *LpxC* inhibitor to be evaluated in phase I clinical trials. Zhao et al. [25] characterized a new inhibitor (LPC-233) of *LpxC*, which can specifically inhibit lipid A synthesis. This slow, tight-binding *LpxC* inhibitor contains a difluoromethyl-allo-threonyl hydroxamate head (Table 1) and has shown outstanding antibiotic activity against a wide range of Gram-negative bacteria, including MDR/XDR *P. aeruginosa* (MIC_90_ = 1 μg/mL). This oral compound is bioavailable and efficiently eliminates infections caused by susceptible and MDR Gram-negative bacterial pathogens in various murine models. In addition, it displays exceptional in vitro and in vivo safety profiles, and no cardiovascular toxicity is detected *in vivo*. These results establish the feasibility of developing oral *LpxC*-targeting antibiotics for clinical use.

LPS biosynthesis requires both UDP-N-acetylglucosamine (UDP-GlcNAc) and acyl-ACP molecules. Both are also necessary for the biosynthesis of peptidoglycan (PG) and phospholipid, respectively, though LPS and PG have individual synthesis pathways [26]. The coordination between these essential surface layers of the OM has not been made clear. It was not until recently that the discovery of a regulatory interaction between the dedicated enzyme involved in the PG and LPS synthesis pathways in *P. aeruginosa* was made. Hummels et al. [27] found that the PG synthesis enzyme *MurA* interacted directly and specifically with *LpxC*. *P. aeruginosa* treated with the *MurA* inhibitor fosfomycin had a PG synthesis inhibition phenotype involving membrane bleb formation and lysis; however, *MurA-*depleted *P. aeruginosa* instead adopted an enlarged, ovoid shape. Moreover, *MurA* was shown to stimulate *LpxC* activity in cells and in a purified system [27,28]. *MurA* is the target of antibiotic fosfomycin, and *LpxC* is an attractive target for developing antibacterial agents against Gram-negative bacteria [17,23]. These imply a potential to develop dual-target drugs that alter both *MurA* and *LpxC* activities while simultaneously disrupting PG and OM assembly. Thereby, the combinatorial antibiotic-inhibiting treatment will be more effective in killing *P. aeruginosa* and/or sensitizing it to other antibiotics that were made ineffective by the barrier function of its envelope.

### 2.2. LPS Modification

*P. aeruginosa* has intrinsic, acquired, and adaptive resistance to a variety of antimicrobials. Antimicrobials are becoming increasingly ineffective, as MDR spreads quickly, leading to infections that are difficult and sometimes impossible to treat. Polymyxins have been revived as the last-line defense against infections by MDR Gram-negative bacteria. Colistin (polymyxin E) targets LPS through the modification of lipid A, which competitively interacts with the anionic lipid A, displacing divalent Ca^2+^ and Mg^2+^ ions to destabilize the OM, subsequently disrupting the IM, leading to cell death [29]. Colistin resistance in Gram-negative bacteria can be chromosomal or plasmidmediated. The first case of plasmid-mediated colistin resistance conferred by the *mcr-1* gene in *Enterobacteriaceae* was reported in 2016. It was observed that *P. aeruginosa* transformed with *mcr-1* demonstrated elevated colistin MIC from 0.5 mg/L to 8 mg/L and polymyxin B MIC from 0.5 mg/L to 4 mg/L [30], suggesting the potential emergence and spread of plasmid-borne colistin resistance threating human health. Most common mechanisms conferring resistance to colistin are directed against modifications of the lipid A moiety of LPS with the addition of positively charged moieties, such as phosphoethanolamine (pEtN) and 4-amino-4-deoxy-L-arabinose (L-Ara4N) (Figure 1), resulting in a reduction in colistin affinity [31,32]. Most Gram-negative bacteria utilize lipid A modifications to evade the host immune response [33]. Changes in *P. aeruginosa* growth conditions can induce the extensive remodeling of lipid A, including the addition or removal of phosphate groups and acyl chains [34]. *P. aeruginosa* clinical isolates showed that polymyxin resistance is associated with the addition of L-Ara4N to phosphate groups within the lipid A and core oligosaccharide moieties of LPS. This pathway refers to the aminoarabinosylation of lipid A [35,36], which a critical prerequisite for the acquisition of colistin resistance in *P. aeruginosa*. Researchers identified a promising inhibitor of the enzyme responsible for colistinresistancemediating lipid A aminoarabinosylation in *P. aeruginosa*. In addition, *ent*-beyerane skeleton (Table 1) may hold promise for the further development of colistin resistance inhibitors [37].

### 2.3. LPS Transport

LPS synthesis and transport pathways are attractive targets for the development of new antimicrobial therapeutics. LPS transport requires MsbA and Lpt proteins. MsbA is a member of the ABC transporter superfamily and performs the first step of LPS transport by flipping core LPS across the IM [38]. LPS is then transported to the cell surface via the Lpt pathway [39]. Two inhibitors targeting MsbA are reported. Inhibitor G907, a quinolone-like compound, binds to the transmembrane pocket of MsbA and locks the protein in an inward-facing LPS-bound conformation [40]. However, this inhibitor is less effective in *P. aeruginosa*. Another MsbA inhibitor, tetrahydrobenzothiopene, stimulates the ATPase activity of MsbA and causes the decoupling of ATP hydrolysis from LPS [41]. Recently, a study on tetrahydrobenzothiopene derivatives (Table 1) was reported. The in vitro evaluation showed that most of the target compounds exhibited a great potency in inhibiting the growth of bacteria. One candidate showed MIC values of 1.1 μM against *E. coli*, 1.0 μM against *P. aeruginosa*, 0.5 μM against *Salmonella*, and 1.1 μM against *S. aureus*. This demonstrates a promising lead compound for the development of new antimicrobial agents against MDR and persistent bacteria [42,43].

**Table 1 antibiotics-13-00119-t001:** Compounds targeting LPS in *P. aeruginosa*.

	Name	Structure	Mechanism of Action	Stage of Development	References
*LPS Biosynthesis*					
1	CHIR-090	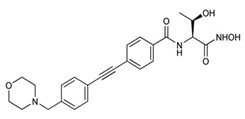	Slow, tight-binding *LpxC*	pre-clinical development	[18,22]
2	ACHN-975	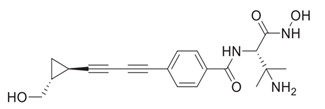	Inhibits *LpxC*	Clinical phase I trial, terminateded due to inflammation	[19,20]
3	PF-5081090	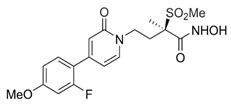	Inhibits *LpxC*	pre-clinical development	[21,22]
4	LPC-233	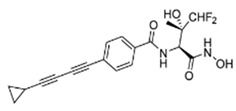	Slow, tight-binding *LpxC*	pre-clinical development	[25]
*LPS Modification*					
5	ent-beyerane skeleton	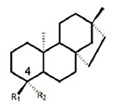	ArnT inhibitor, potential colistin resistance inhibitor	prepclinical discovery	[37]
*LPS Transport*					
6	G907	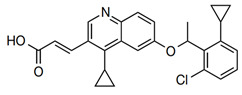	Inhibits MsbA	pre-clinical development	[40]
7	Tetrahydrobenzothiophene derivatives	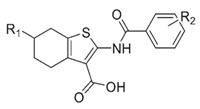	Inhibits MsbA	pre-pclinical discovery	[42,43]
8	Novobiocin	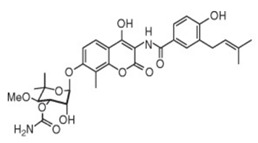	Inhibits MsbA, enhances ATPase of LptB	pre-clinical development	[44]
9	Murepavadin, POL7080	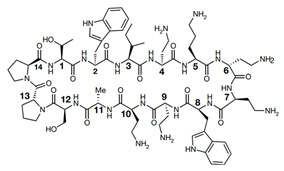	Inhibits LptD	Clinical phase III trial, terminateded due to kidney problem	[45]

Recently, the biological effects of verapamil, an inhibitor of ABC transporters, were investigated in *P. aeruginosa*. It was noticed that IC_50_ for novobiocin decreased 34% in the presence of verapamil [44]. Verapamil did not inhibit *P. aeruginosa* growth but increased its sensitivity to novobiocin. The molecular modulization of protein MsbA followed by a docking analysis revealed that novobiocin and verapamil interacted at a common site on MsbA protein. The result indicates that both novobiocin and verapamil act as MsbA potential competitive inhibitors.

Lpt proteins represent another promising target for developing new classes of antibiotics. Murepavadin (POL0780) is a peptidomimetic antibiotic that interacts with the LPS transporter LptD to block LPS assembly and insertion into the OM [45] (Figure 1 and Table 1). As the first OM protein-targeting antibiotic to enter late-stage clinical development, murepavadin displays both potent activities in vitro against *P. aeruginosa*, including MDR clinical isolates with MIC_90_ at 0.12 mg/L, and in vivo pharmacokinetics assays in mouse models of infection. Regrettably, the phase III trial, testing murepavadin in 150 enrolled patients with hospital-acquired pneumonia who required mechanical respiratory support, was halted due to safety concerns. Fifty-six percent of patients reported kidney problems in one of the trials [46].

It is known that colistin kills *P. aeruginosa* through the modification of lipid A on the OM. However, how it disrupts the inner membrane is not clear. Sabnis et al. [47] designed a new therapeutic approach and exposed *P. aeruginosa* to colistin and/or murepavadin. An MIC assay showed that murepavadin sensitized *P. aeruginosa* to colistin by increasing LPS abundance in the IM. In addition, in both in vitro clinical MDR isolates and an in vivo mouse model of lung infection, the combination therapy of colistin and murepavadin demonstrated enhanced efficacy in the killing and clearance of *P. aeruginosa*. These results also demonstrated that colistin exerts bactericidal activity by targeting LPS in the IM. A recent study demonstrated that murepavadin impaired bacterial OM integrity, which induced the envelope stress response in *P. aeruginosa*. The combination of murepavadin with ceftazidime/avibactam slowed down the resistance development in a mouse model of acute pneumonia by *P. aeruginosa* infection [48]. Taken together, these experiments provide robust evidence that colistin targets LPS directly and can be used in a combination of murepavadin and other antibiotics for the treatment of serious infections caused by *P. aeruginosa*.

### 2.4. Porins

The molecular characterization of the OM is essential for understanding how antibiotics penetrate this barrier, both for the development of new therapeutic strategies and for rational drug design. Porins are beta barrel pore proteins contained in the OM of Gram-negative bacteria. These proteins possess an internal hydrophilic channel that generally restricts the entry of most lipophilic molecules and only permits the passage of certain small hydrophilic molecules from the external environment to the interior cell [49,50]. A study on antibiotic properties using high-throughput screening of more than 8 million compounds targeting Gram-negative infections confirmed that the successful compounds were mostly hydrophilic small molecules. This points toward size exclusion by porins [51,52]. In contrast to Enterobacteria, *Pseudomonas* lack large diffusion porins, such as OmpF and OmpC [53]. The OM permeability of *P. aeruginosa* was 100-fold lower than *E. coli* [54]. In addition, the low permeability in *P. aeruginosa* is attributed to a large family of OprD (also termed OccD1) porins comprising 19 members. These porins act together as specific channel proteins involved in the uptake of different nutrients [50,55,56,57].

OprF is the most abundant non-lipoprotein on the OM of *P. aeruginosa*. Owing to its C-terminal containing a PG-binding domain, OprF is mainly involved in maintaining the OM structure [58]. Mutations on OprF confirmed that its N-terminal is responsible for protein production and membrane insertion, while its C-terminal is liable for stable interaction with PG anchoring on the OM [59]. As OprF and OprI are highly conserved and induce a cross protective immunity across all *P. aeruginosa* strains, they become promising vaccine candidates for the control of *P. aeruginosa* infection. A phase III clinical trial of IC43, a hybrid OprF/I vaccine, has been completed [60]. OprF involves adhesion, biofilm formation, and virulence. In comparison to the wild-type, isogenic OprF mutant, and an OprF-complement strain, OprF is required for *P. aeruginosa* virulence as the OprF mutant strain displays reduced cytotoxicity in cells [61]. A study on the OprF mutant in a vertebrate model showed that OprF protected *P. aeruginosa* against macrophage clearance by avoiding bacterial elimination in acidified phagosomes [62]. Using scanning electron microscopy, a comparative study was conducted on biofilm formation, examining both a wild-type and an isogenic OprF mutant of *P. aeruginosa*. The results showed that OprF played a dynamic role in *P. aeruginosa* virulence. The absence of OprF resulted in a slow growth rate corresponding to an elongated lag phase and reduced biofilm production [63].

The high stability of *P. aeruginosa* OM is due to the presence of the OprH, the smallest porin found in *P. aeruginosa*. Edrington et al. [64] provided the first molecular structure of OprH and evidence for multiple interactions between OprH and LPS that likely contributed to the antibiotic resistance of *P. aeruginosa*. Lee et al. [65] built various simulation systems to investigate the impact of different LPS molecules on OprH structure and dynamics. The results showed that the OprHLPS interactions mainly depended on the secondary structure of OprH and the chemical structure of LPS, which may contribute to bacterial AMR. Similar results were validated using a solution nuclear magnetic resonance spectroscopy system [66]. As polymyxin-resistant *P. aeruginosa* clinical isolates are capable of lipid A modification, OprH is regarded as a potential target for novel antimicrobial therapies.

Carbapenem is a mainstay therapy for *P. aeruginosa* infection. In general, carbapenems can efficiently cross the OM by passing through the aqueous channels, such as OprD in *P. aeruginosa*. Reduced permeability caused by downregulated OprD protein appears to be the most common mechanism of intrinsic resistance to carbapenem [67]. Mutations in OprD [68,69,70,71,72] are associated with imipenem resistance and reduced susceptibility to meropenem through the loss of or change in OprD. One study investigated OprD mutations in locally prevalent MDR *P. aeruginosa* strains in cystic fibrosis of clinical relevance. An analysis of whole-genome sequencing found shared strain sub-lineages with specific OprD variants pre-existing in the local population before spreading between patients [73].

Wolter et al. [74] were the first to report carbapenem resistance resulting from the insertional inactivation of the *oprD* gene by insertion sequence elements in seven clinical *P. aeruginosa* isolates. These isolates exhibited dual resistance to fluoroquinolones and imipenem. In an isogenic pair of MDR *P. aeruginosa* clinical isolates, we uncovered an extra m6A methylation in the promoter of an endotoxinAregulating gene, *toxR*, most likely causing the higher expression of OpdQ, a member of the OprD porin family. We proposed an epigenetic regulation of *opdQ* expression pertinent to the phenotypic change in *P. aeruginosa* from resistant to susceptible to piperacillin/tazobactam and increases in MIC to meropenem [75].

## 3. New Strategy Development against *P. aeruginosa* Infection

New antibiotics are considered ‘drugs of last resort’ against MDR bacteria. The delay in discovering new antibiotics has exacerbated the MDR problem and leaves a gap between the diagnosis of an MDR pathogen and effective treatment. Alternative therapies, especially those differing from traditional antibiotics, are emerging. These non-traditional therapies in the global preclinical antibacterial pipeline are mainly phage therapies, anti-virulence therapies, antibodies (antibody–drug conjugates), vaccines, and nanoparticles [76,77]. Compared to traditional antibiotics, these new methods have the advantage of avoiding negative effects on host commensal bacteria and AMR development.

### 3.1. Phages

Bacteriophage therapy is one of the promising alternatives against MDR *P. aeruginosa*. Many research studies have demonstrated the ability of phages to eradicate *P. aeruginosa* [78,79,80]. Several clinical studies have been conducted to evaluate the effectiveness of phage therapy in treating specific *P. aeruginosa* infections (https://clinicaltrials.gov) (accessed on 20 December 2023).

A randomized phase 1/2 trial (NCT02116010) was designed to compare the efficacy and tolerability of a cocktail of 12 natural lytic bacteriophages (PP1131) with standard of care for patients with burns. The study showed that PP1131 decreased the bacterial burden in burn wounds at a slower pace than standard of care at low concentrations of phage. It was felt that a higher phage dose in a larger number of participants was warranted for further studies [81].

Trials of AP-PA02, another cocktail of bacteriophages designed to fight *P. aeruginosa* infections in patients with chronic pulmonary infection and CF bronchiectasis (NCT04596319, phase I/II) and non-CF bronchiectasis (NCT05616221, phase II), are underway. Studies aim to evaluate the safety, tolerability, phage kinetics, and efficacy of inhaled AP-PA02.

Another two clinical trials are being conducted: NCT04323475, a phage cocktail-SPK therapy for second-degree burn wounds in adult patients, and NCT04684641, a bacteriophage therapy YPT-01 for *P. aeruginosa* infections in adults with CF. To date, no new participants have yet been recruited or enrolled.

Phage therapy has so far failed to translate into positive outcomes in the limited number of clinical trials that have been performed. One problem is the rapid evolution of phage resistance, which limits the clinical efficacy of phage therapy. Yang et al. [82] strategically formulated a cocktail of phages that successfully suppressed the evolution of resistance. After prolonged incubation, phage resistance in *P. aeruginosa* was observed. Lipid remodeling during phage infection may alter binding and subsequent infection dynamics [83]. One case study described phage therapy for a complex bone and joint infection of XDR *P. aeruginosa*, in which phage therapy combined with ceftolozane/tazobactam and colistin resulted in rapid wound healing over 2 weeks [84]. Another recent case reported a successful aerosolized bacteriophage with concomitant antibiotic treatment of a chronic lung infection due to MDR *P. aeruginosa*. The patient showed clinically significant improvement even without the complete eradication of *P. aeruginosa* lung colonization [85]. Therefore, effective phage therapy strategies including phageantibiotic synergies and optimizing phage administration will likely improve the outcome in future trials [86].

### 3.2. Vaccines

Extensive research has focused on vaccine development against *P. aeruginosa* over the last 50 years. Four vaccines have entered phase III trials during these years. While some showed promising results, no anti-*P. aeruginosa* vaccine has yet been approved. LPS and OM proteins are important antigens of *P. aeruginosa*, which have been shown to be immunogenic for hosts. However, due to the high diversity of *P. aeruginosa* serotypes, it is hard to design a vaccine effective for all serotypes [87].

Octavalent O-polysaccharide-exotoxin A conjugate (Aerugen^®^) is an LPS-based conjugate vaccine. A phase I study showed high affinity IgG response to exotoxin and LPS in healthy volunteers. A phase II study initially showed similar immunoglobulin functions in CF patients not colonized with *P. aeruginosa*. Unfortunately, it did not have an impact on clinical outcomes. The phase III trial in CF patients was finally stopped as the interim results did not show significant differences between the placebo and control groups [88].

Another vaccine that has reached phase III clinical trial is His-tagged OprF-OprI fusion protein (IC43). This vaccine seems to be an effective candidate due to its better safety and immunogenicity profile in phase I and II trials [89,90]. In phase III, 800 patients on mechanical ventilation were randomized to either IC43 100 μg or saline placebo, given in two vaccinations 7 days apart. The mortality in patients twenty-eight days after the first vaccination and the immunogenicity and safety of IC43 were evaluated. The IC43 vaccine was well tolerated and achieved high immunogenicity in these patients, but the overall mortality rate in the vaccinated patients was not significantly different from that in the placebo group [60].

The U.S. Food and Drug Administration licensed two new pneumococcal conjugate polyvalent vaccines for the prevention of invasive pneumococcal disease in 2021 [91]. It inspired the development of a trivalent vaccine that targets multiple antigens affecting major virulence factors in *P. aeruginosa*. Recently, a recombinant protein POmT comprising three antigens—the full-length V antigen (PcrV) of the *P. aeruginosa* type III secretion system (TTSS), the OM domain of OprF, and a non-catalytic mutant of the carboxyl domain of exotoxin A—was created. A significant improvement in acute lung injury and a reduction in the acute mortality of *P. aeruginosa*-induced acute lung injury were observed in a mouse model [92].

The great success of mRNA-based COVID-19 vaccines during the pandemic attracted more researchers to employ the potential of mRNA as a preventive and therapeutic vaccine for distinct infectious diseases. Compared to conventional vaccines, mRNA vaccines have a favorable safety profile and higher efficiency because mRNA is non-infectious and poses little concern for DNA integration. PcrV protein at the tip of TTSS is an effective target for active and passive immunization against *P. aeruginosa* [93,94,95]. Recently, researchers reported the development of mRNA vaccines targeting the TTSS of *P. aeruginosa*. Mice were vaccinated with nucleoside-modified mRNA encapsulated in lipid nanoparticles. Together, the mRNA-immunized mice showed improved survival, decreased lung bacterial loads, and fewer pathological changes in the lung compared to saline controls [96].

Meanwhile, another group of researchers designed two mRNA vaccines targeting PcrV and fusion protein OprF-I, which demonstrated highly immunogenic antigens conserved across different serotypes of *P. aeruginosa*. Notably, both PcrV and OprF-I mRNA vaccines could induce immunity; the PcrV vaccination elicited a higher level of humoral and cellular immune responses and significantly reduced the colonization of *P. aeruginosa* in the skin, lung, liver, spleen, and kidney, providing a broad protective effect against *P. aeruginosa*. Furthermore, the immune response and protection elicited by a single mRNA vaccine and combined ones showed the superiority of mRNA vaccines over PcrV and OprF protein vaccines, respectively [97]. Many Gram-negative bacteria share highly homologous TTSS. mRNA targeting different bacteria can be designed quickly and scaled up with low cost. This new approach can provide a potential application prospective for different Gram-negative pathogens.

### 3.3. Nanoparticles

Photothermally active nanomaterials are emerging as potent antimicrobial agents [98]. Recently, a selective photothermal therapy based on LPS aptamer functionalized nanorods for MDR *P. aeruginosa* infection was reported [99]. Animal experiments showed that the nanorods permitted the active targeting of LPS on the surface of Gram-negative bacteria and a specific anti-inflammatory ability in the MDR *P. aeruginosa*-infected wound murine model. In addition, the nanorods can precisely overcome MDR *P. aeruginosa* through physical damage and effectively reduce excess M1 inflammatory macrophages to accelerate the healing of infected wounds. Overall, this molecular therapeutic strategy displayed great potential as a prospective antimicrobial treatment for MDR *P. aeruginosa* infection.

The combination of bacteria-imprinting technology and photothermal therapy has emerged as a potential therapeutic strategy for fighting drug-resistant bacteria [100,101,102,103]. A group of scientists reported a promising material: photothermal molecularly imprinted polymers (PMIPs) [104]. Based on the affinity of *P. aeruginosa* LPS with boric acid, LPS-imprinted PMIPs were synthesized for the study of the efficient capture and elimination of *P. aeruginosa*. Fluorescent images demonstrated that the engineered PMIP had low toxicity to normal cells, higher affinity to LPS, and more significant targeting capability toward *P. aeruginosa* than nonimprinted polymers. Although PMIP alone showed low anti-biofilm activity against *P. aeruginosa* in established biofilms, most viable cells were effectively eliminated by the combination of PMIP and irradiation near-infrared light therapy. So, this new technology can be used as a potentially powerful tool for the safe and efficient deactivation and removal of *P. aeruginosa*.

## 4. Challenges

The rapid growth of AMR/MDR is driven by the misuse and overuse of antibiotics that are commonly and widely used to treat bacterial infections. Antibiotics can wipe out harmful pathogens of concern and save lives. Simultaneously, antibiotics eradicate beneficial microbes, reduce microbiota diversity, and alter metabolic activity with deleterious consequences for human health [105]. Additionally, antibiotics may select bacteria with AMR to overgrow as these bacteria evolve and respond to the selective pressures placed upon them, whereas the combinatorial use of antibiotics can lead to the production and spread of MDR bacteria. Moreover, the source of antibiotics has probably been exhausted and the development of new antibiotics has become slow and expensive [106], resulting in the AMR crisis that we are experiencing now. Developing and implementing novel therapeutic strategies other than traditional antibiotic therapy is thus imperative in combating AMR. Here, we reviewed the recent advances in the development of novel therapeutic strategies against MDR *P. aeruginosa*.

*P. aeruginosa* is among the largest of the bacterial genomes with its genome size at a range of 5.5–7.0 million base pairs. This large genome facilitates *P. aeruginosa*’s evolutionary adaptation to diverse environments and development of resistance to antibiotics, phages, and/or vaccines. Additionally, *P. aeruginosa* possesses an abundance and diversity of bacteriophages, both lysogenic and lytic, driving bacterial evolution and providing a great source of phage candidates for phage therapy. However, few of these natural phages have been characterized in detail regarding their target specificity and toxic contents, leading to concerns about the safety, reliability, and efficacy of their applications. Recently, an increasing number of MDR *P. aeruginosa* clinical isolates have been whole-genome sequenced, revealing its high genome diversity, dynamic evolution, and MDR complexity. This diversity and complexity makes the prevention and treatment of MDR *P. aeruginosa* much more challenging. Notably, PAO1 and PA14 are the most commonly employed reference strains with moderate and hyper-virulent phenotypes, respectively. However, sequencing has demonstrated significant deviations from the clinical isolates of *P. aeruginosa* infection [107]. In the genomic analysis of four isolates from different environments and CF patients, these isolates had a defective Las quorum-sensing system, but remained virulent when compared to PAO1 and PA14 [108]. This result reminds us that special care should be taken when translating laboratory results into clinical applications, and whole-genome sequencing should be used to reduce variations in the accrued data in *P. aeruginosa* studies [109].

Most of the aforementioned novel therapeutic strategies are target-specific, making them attractive alternatives to antibiotic treatment in maintaining healthy microbiota diversity. Owing to our limited knowledge of phage-targeting specificity to prevent bacterial development of anti-phage systems [110], phage cocktails, instead of individual phages, are usually designed for phage therapy. A broad-range bacteriophage cocktail was found to be superior to the individual phage in destroying *P. aeruginosa* biofilm and providing a faster treatment in mice with acute respiratory infection [111]. In addition, synergistic phage–antimicrobial cocktail therapy can improve phage therapy effectiveness. This has been reported in several successful personalized phage therapies for patients suffering from chronic MDR *P. aeruginosa* life-threatening infection [84,85,112]. In PAO1 and PA14 test models, the evolution of bacterial resistance to a lytic phage attack changed the efflux pump mechanism, causing increased sensitivity to drugs from several antibiotic classes [113]. To effectively combat MDR *P. aeruginosa*, combinatorial treatment, complementarily or synergistically, may be the better strategy.

## Figures and Tables

**Figure 1 antibiotics-13-00119-f001:**
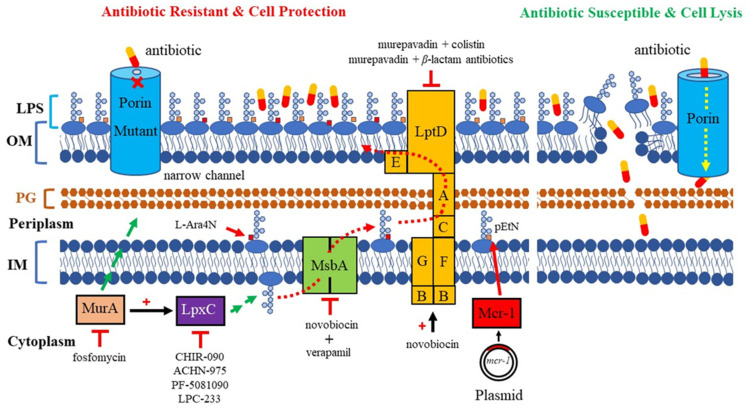
Mechanism of antimicrobial resistance related to LPS and porin in *P. aeruginosa*. Red dashed line arrows: LPS is transported from its synthesis site to the outer membrane. Yellow dashed line arrow: porins are channels that allow the entry of antibiotics. Red cross sign: *narrow-channel* porin is a barrier that blocks antibiotics from passing through the outer membrane. Red arrows: addition of a pEtN or L-Ara4N positively charged moiety to the lipid A of LPS. Red flat-headed arrows: inhibition of pathway. Red plus signs: upregulation of enzymatic activity; *MurA* stimulates *LpxC* activity, and novobiocin directly binds the ATPase LptB and increases the activity of the LPS transporter. *MurA* commits the first step of PG biosynthesis (long green arrow). *LpxC* commits the catalysis step of LPS biosynthesis (short green arrow).

## Data Availability

No new data were created or analyzed in this study. Data sharing is not applicable to this article.

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
