# Peer review of "Advances in Development of Novel Therapeutic Strategies against Multi-Drug Resistant Pseudomonas aeruginosa"

_antibiotics, 2024, doi:10.3390/antibiotics13020119_

Round 1

Reviewer 1 Report

Comments and Suggestions for Authors

good research , good luck.

Reviewer evaluation

Introduction

·        (https://arpsp.cdc.gov/profile/antibiotic-resistance/mdr-pseudomo- nas-aeruginosa).  It’s better to replace it by number and put this in references

·        (https://www.cdc.gov/hai/outbreaks/CRPA-artificialtears.html, accessed on December 1, 2023). same remark

·        The molecular mechanisms of AMR are complex………… of new therapeutic strategies. Added reference(s) to these information

·        In the end of your introduction added clear objective to the present study

2.1. LPS biosynthesis

….involving membrane bleb formation and lysis, however in P. aeruginosa depleted

2.3. LPS Transport

(https://www.swissbiotech.org/listing/polyphor-temporarily-halts-phase- iii-studyor-the-treatment-of-patients-with-nosocomial-pneumonia/). It’s better to replace it by number and put this in references

2.3. LPS Transport

Recently, the biological effects of verapamil, an inhibitor …..verapamil (39). Verapamil didn’t inhibit P. .. competitive inhibitors (39). This paragraph is from the same reference (39), why you write it twice or repeat it

Lpt proteins represent another promising target for developing new classes ……. into the OM (40). As the first OM …….. model of infection (40). The same remark for this paragraph for reference (40).

2.4. Porins

3.1. Phages

(https://clinicaltrials.gov)

https://clinicaltrials.gov/study/NCT04596319)

https://clinicaltrials.gov/study/NCT05616221

(https://clinicaltrials.gov/study/NCT04323475)

(https://clinicaltrials.gov/study/NCT04684641)

It’s better to replace them by number and put them in references

3.2. Vaccines

(https://doi.org/10.1101/2023.06.09.544431) same remark

3.3. Nanoparticles

(https://creativecommons.org/licenses/by/4.0/) same remark

4. Challenges

To effectively combat MDR P. aeruginosa, combinatorial treatment, complementarily  or synergistically, may be the better strategy. Added reference(s) to this idea or remove it

Figure 1. Mechanism of antimicrobial resistance related to LPS and porin in P. aeruginosa this fig need reference(s)

References

Murray, C.J.; Ikuta, K.S.; Sharara, F.; Swetschinski, L.; Aguilar, G.R.; Gray, A.; Han, C.; Bisignano, C.; Rao, P.; Wool, E.; et al. Global burden of bacterial antimicrobial resistance in 2019: a systematic analysis. Lancet. 455 2022;399(10325):629-55 added authors to the first cited reference

Write the name of bacteria species cited in all references of P. aeruginosa or others like E.coli in italic

Reviewer 2 Report

Comments and Suggestions for Authors

The review is very interesting for researchers who wish to develop and contribute to new therapeutic strategies against multiresistant strains of P. aeruginosa, a serious public health problem.

In the manuscript, the authors mention several chemical molecules developed and tested against bacterial strains of P. aeruginosa. It would be important to present the structures of these molecules because it is easier for the reading public to associate some biological properties from the molecular structures. On the other hand, it is recommended to present a table that consolidates the information of the different therapeutic strategies with the results of the activity in order to compare the results related in this review.

Author Response

Thanks!

Reviewer 3 Report

Comments and Suggestions for Authors

Line 9, abstract part, “P. aeruginosa with multi-drug resistance…” or “Multi-drug resistant…”?

Line 20, Novel and ? strategies? What did the authors want to express here? How can they prove “Novel”?

Author Response

Thanks!

Reviewer 4 Report

Comments and Suggestions for Authors

Dear authors,

Here are my suggestions to improve your manuscript:

Line 42 – spp. is not italicized.

Line 97 – the reference Zhao et al must be accomplished by its number, I believe to be 20

Line 112, same for Hummels et al; please check lines 187; 234; 236; 297

Line 112 – please check the MurA gene name is italicized. Please check the others gene’s names

Line 129 – please modify Ca++ and Mg++ to Ca2+ and Mg2+

Line 168 – it is good if you mention the pharmacological class of verapamil due to it is not an antibiotic. Is there interest for other associations between non-antibiotic drugs and antimicrobials? Maybe you kindly include this information.

Line 211 – Enterobacteria is not italicized.

line 258 – why did you use “we”? you are telling that Wolter and colleagues have discussed. If it is your opinion, please make it clear using for example…in addition…; according to their findings…or change it to “they”

line 403 – P. aeruginosa to have a wide genome also implies to keep stable, making wild and clinical strains differ in phenotypes but conserve important parts in the genome, which may be a target in further therapies. Please check this idea and include in this section.

Final consideration: it should be presented a summary of some perspectives. You have shown the challenges but which methodologies to be possible? Anti-quorum sensing; use of non-antimicrobials in association with antibiotics; natural bioactive molecules (this topic is missing in my opinion); biofilm eradication; is pyocyanin involved in LPS formation? You are free to decide if these final considerations are important to improve your manuscript. Hope success to your journey.

Author Response

Thanks!
